# Integrative Analysis of N6-Methyladenosine-Related Enhancer RNAs Identifies Distinct Prognosis and Tumor Immune Micro-Environment Patterns in Head and Neck Squamous Cell Carcinoma

**DOI:** 10.3390/cancers14194657

**Published:** 2022-09-25

**Authors:** Hongshi Cai, Jianfeng Liang, Yaoqi Jiang, Rukeng Tan, Chen Hou, Jinsong Hou

**Affiliations:** 1Department of Oral and Maxillofacial Surgery, Hospital of Stomatology, Guanghua School of Stomatology, Sun Yat-sen University, Guangzhou 510055, China; 2Guangdong Provincial Key Laboratory of Stomatology, Guanghua School of Stomatology, Sun Yat-sen University, Guangzhou 510080, China

**Keywords:** head and neck squamous cell carcinoma, N6-methyladenosine, enhancer RNA, immunity

## Abstract

**Simple Summary:**

Head and neck squamous cell carcinoma (HNSCC) has high morbidity and mortality. The interaction between immune cells and tumor cells in the tumor micro-environment is an important factor affecting the tumor progression and prognosis of HNSCC patients. More biomarkers and targets need to be explored to improve patient outcomes. The m^6^A modification on enhancer RNAs (eRNAs) is associated with the signature of active enhancer, and the function of m^6^A driving eRNAs in tumor progression has not been reported. In this study, we screened and identified a risk model containing 5 m^6^A-related eRNA, which can better predict the survival and immunotherapy outcome of patients. The role of m^6^A-related eRNA in HNSCC cells was verified in vitro. We also combined the risk score and multiple clinical features to construct a nomogram for predicting OS of HNSCC patients, which provides an effective quantitative analysis tool for guiding the personalized precise treatment for patients.

**Abstract:**

At present, the prognostic value of N6-methyladenosine (m6A)-related enhancer RNAs (eRNAs) for head and neck squamous cell carcinoma (HNSCC) still remains unclear. Our study aims to explore the prognostic value of m6A-related eRNAs in HNSCC patients and their potential significance in immune infiltration and immunotherapy. We constructed a 5 m6A-related eRNAs risk model from The Cancer Genome Atlas (TCGA) HNSCC dataset, using univariate and multivariate Cox and least absolute shrinkage and selection operator (LASSO) regression analysis. Based on the SRAMP website and in vitro experiments, it was verified that these 5 m6A-related eRNAs had m6A sites, the expression of which was regulated by corresponding m6A regulators. Moreover, we constructed a nomogram base on 5 m^6^A-related eRNAs and confirmed the consistency and robustness of an internal TCGA testing set. Further analysis found that the risk score was positively associated with low overall survival (OS), tumor cell metastasis, metabolic reprogramming, low immune surveillance, lower expression of immune-related genes, and higher expression of targeted genes. Finally, we verified that silencing MIR4435-2HG inhibited HNSCC cell migration and invasion. This study contributes to the understanding of the characteristics of m6A-related eRNAs in HNSCC and provides a reference for effective immunotherapy and targeted therapy.

## 1. Introduction

Head and neck squamous cell carcinoma (HNSCC) occurs mainly on the mucosal surfaces of the oral cavity, pharynx, and larynx. The occurrence of HNSCC is usually associated with risk factors such as smoking, betel nut chewing, alcoholism, and human papillomavirus (HPV) infection, resulting in the inactivation of tumor suppressor genes and overexpression of oncogenes, which promote tumor cell proliferation and distant metastasis [1]. According to the estimated data on global cancer incidence and mortality in 2020 published by the International Agency for Research on Cancer, HNSCC is the eighth major malignant tumor, with 878,348 new diagnoses of HNSCC and 444,347 new deaths [2]. The National Cancer Center of China reported 77,900 new HNSCC cases (approximately 5.64/10^5^) and 40,100 deaths (approximately 2.90/10^5^) in 2016 [3]. Despite great progress in the diagnosis and treatment of HNSCC, more than 60% HNSCC patients are initially diagnosed as stage III or IV disease, with obvious local infiltration, cervical lymph node metastasis, high risk of local recurrence, and distant metastasis, leading to poor prognosis of HNSCC patients [2,4]. HNSCC patients are treated in a multidisciplinary manner, including a combination of surgery, chemotherapy, and radiotherapy. Applying immune checkpoint inhibitors is also a suitable choice for patients with unresectable recurrent or metastatic HNSCC. In the past decade, the application of high-throughput sequencing data to search for prognostic biomarkers for HNSCC and its integration with immunology have provided new opportunities for therapeutic intervention.

N6-methyladenosine (m^6^A) is the most abundant post-transcriptional modification in eukaryotic RNA and widely exists in eukaryotic messengers RNA (mRNAs) and long non-coding RNAs (LncRNAs), microRNA, circular RNA, tRNA, rRNA, and snRNA [5,6,7,8]. Similar to DNA and histone methylation, m^6^A RNA methylation is also dynamically reversible. S-adenosylmethionine (SAM) is the methyl donor of m^6^A. m^6^A is formed under the catalysis of methyltransferase complexes (methyltransferase complex, MTC) composed of catalytic subunits (METTL3) and other auxiliary subunits (METTL14, WTAP, VIRMA, ZC3H13, RBM15, RBM15B, CBLL1, and METTL16) [9,10,11,12,13,14,15,16]. Demethyltransferase FTO and ALKBH5 catalyze the removal of m^6^A [17,18]. m^6^A modifications are recognized by m^6^A-binding proteins and mediate their biological functions. m^6^A is recognized by m^6^A-binding proteins, which mediate its biological function in vivo. At present, the functional binding proteins of m^6^A are mainly proteins containing YTH domain (YTHDF1, YTHDF2, YTHDF3, YTHDC1, and YTHDC2) [19,20,21,22,23,24], HNRNPA2B1 [25], HNRNPC [26], eIF3A [27], IGF2BP1/2/3 [28], RBMX [29], FMR1 [30], ELAVL1 [31], and LRPPRC [32]. In recent years, with the study of methylation and demethylation of m^6^A on RNA in tumors, it has been found that m^6^A binding proteins recognize m^6^A to regulate RNA splicing, nuclear export, stabilization, and translation, which further affect the proliferation, differentiation, invasion, metastasis, immune escape and stem cell maintenance of various tumors [33].

With the wide application of high-throughput sequencing data, the complexity and diversity of transcriptomes are gradually recognized. Recently, a kind of enhancer RNA (eRNAs) transcribed from the enhancer region has been found, which belongs to non-coding RNA and is an active marker of enhancer, participating in and assisting the transcription process in gene regulation [34,35]. Enhancers control and maintain cell characteristics, and eRNAs generated by their transcription play a functional role in gene regulation. ERNAs are expressed in a variety of tumor tissues and have important tissue specificity [36]. McCleland et al. defined an eRNA transcribed from the c-MYC enhancer region by integrating transcriptomic and chromatin immunoprecipitation sequencing (ChIP-seq) analysis, that is, colon cancer-associated transcript 1 (CCAT1). CCAT1 can regulate the expression of cMYC, and the overexpression of CCAT1L in a variety of tumor types increases the expression of cMYC and promotes tumor growth [37]. Since eRNA has tissue specificity and is involved in the regulation of cell fate, eRNA may become a potential target for disease diagnosis and treatment [38]. Using the methylation-inscribed nascent transcripts sequencing (MINT-seq) method, Lee et al. found that a large number of selective m^6^A deposits were generated on the nascent eRNAs. The eRNAs modified by m6A represented highly active enhancers and played an important role in promoting transcriptional activation [39]. However, the mechanism of eRNA regulating tumor cell development through m6A has not been reported. With the continuous development of bioinformatics, we believe that understanding the role of m6A-related eRNAs in HNSCC can help discover new biomarkers and therapeutic targets.

This study explored the prognostic value of m^6^A-related eRNAs in HNSCC patients and its potential significance in immune infiltration and immunotherapy. Using The Cancer Genome Atlas (TCGA)-HNSCC dataset, 5 m^6^A-related eRNAs were identified by univariate Cox regression analysis, least absolute shrinkage and selection operator (LASSO) Cox regression, and multivariate Cox regression analysis and verified by in vitro experiments. Then, a risk model was constructed to predict the prognosis of patients with HNSCC. For further evaluation of the enriched biological functions and comparison between high- and low-risk groups in immune cell infiltration, gene set enrichment analysis (GSEA) and weighted gene coexpression network analysis (WGCNA) were performed on the risk model. The expression of human leukocyte antigen, immunotherapy markers, and targeted therapy target genes in high- and low-risk groups were compared and analyzed to predict the therapeutic effect. Finally, the effect of eRNA MIR4435-2HG on the migration and invasion of head and neck squamous cell carcinoma cells was verified by in vitro experiments.

## 2. Materials and Methods

### 2.1. HNSCC Samples and Cell Culture

With the approval of the Stomatology Hospital Research Ethics Committee, twenty HNSCC samples and paired adjacent non-cancerous normal tissues (ANNT) were acquired from the Stomatology Hospital of Sun Yat-sen University. Informed consent was signed by each patient. The human HNSCC cell lines SCC25 and CAL27 were purchased from the American Type Culture Collection (ATCC). DMEM/F12 (Gibco, New York, NY, USA) medium containing 10% fetal bovine serum (FBS, WISENT, Montreal, QC, Canada) and 400 ng/mL hydrocortisone (MACKLIN, Shanghai, China) was used to culture SCC25 cells. In DMEM (Gibco, USA) medium containing 10% FBS, CAL27 cells were grown. Cells were cultured at 37 °C in a humidified 5% CO_2_ incubator.

### 2.2. Data Collection, Processing, and Correlation Analysis

RNA-seq data, clinical data, follow-up data, and tumor mutation load data of HNSCC patients in TCGA (including 502 tumor samples and 44 normal controls) were downloaded from UCSC xena. Convert RNA-seq data to a log2 (TPM + 1) value for further analysis. Based on the eRNA annotation files, 1580 eRNA expression matrices were extracted from the TCGA dataset. Three cases of head and neck squamous cell carcinoma without survival time and state were removed, and the “caret” package of R software (version 4.1.0) was used to randomly and evenly divide the data set of 501 patients with head and neck squamous cell carcinoma (with follow-up data) into a training set (353 cases) and test set (148 cases) according to the ratio of 7:3. In addition, the Cancer Immunome Atlas (TCIA, https://www.tcia.at, accessed on 26 November, 2021) was used to download the immunotherapy score data for patients with head and neck squamous cell carcinoma.

### 2.3. Identification of Prognostic m6A-Related eRNAs

The absolute value of the Pearson correlation coefficient with m^6^A regulator > 0.4 and *p* ≤ 0.001 were considered as cutoff values for defining m^6^A-related eRNA, and their coexpression network was plotted. In the TCGA-HNSCC cohort, multivariate Cox regression analysis was used to identify eRNAs with statistically significant overall survival (OS). Then, LASSO Cox regression analysis was used to screen out potential prognostic m6A-related eRNAs. Finally, 5 independent prognostic m^6^A-related eRNAs and corresponding coefficients were identified by multivariate Cox proportional hazard regression analysis, which mainly used the “survival” package and “glmnet” package [40].

### 2.4. Construction and Validation of the Risk Model Based on Prognostic m^6^A-Related eRNAs

According to the expression of 5 m^6^A-related eRNAs in each sample and the corresponding coefficient, the risk score of each sample was calculated. The risk score of each sample was calculated as follows: risk score = expression of AC005562.1 × coefficient of AC005562.1 + expression of COPDA1 × coefficient of COPDA1 + expression of LINC00271 × coefficient of LINC00271 + expression of MIAT × coefficient of MIAT + expression of MIR4435-2HG × coefficient of MIR4435-2HG [41]. Using the median risk score as the cutoff value, we separated the training set into high-risk and low-risk groups. The Kaplan–Meier survival curve was generated to assess the survival difference between the two groups, which was further evaluated by the log-rank test. With the “SurvivalROC” package, the risk model’s prediction accuracy was further evaluated. The receiver operating characteristic (ROC) curves for the 1-, 3-, and 5-year OS of the TCGA-HNSCC training set were constructed, and the area under the curve (AUC) values were calculated. Furthermore, the effectiveness and reliability of the risk model in the TCGA-HNSCC testing set were assessed simultaneously using survival analysis and ROC analysis.

### 2.5. Construction and Validation of Nomogram

The prognostic value of clinical feathers in the TCGA-HNSCC cohort, including age, sex, stage, grade, alcohol, and risk score, was evaluated using multivariate COX regression. Variables with *p* < 0.10 were used to construct nomograms in the training set through the r package “rms”. The calibration curve was drawn to evaluate the consistency between actual survival and predicted survival. ROC and C-index were drawn to assess the prediction accuracy of the line chart. Finally, the nomogram was internally verified using the same method on the TCGA-HNSCC testing set.

### 2.6. Gene Set Enrichment Analysis

GSEA was performed on both high- and low-risk groups using the “limma” package and the “clusterProfiler” package [42,43]. Hallmark gene sets and Kyoto Encyclopedia of Genes and Genomes (KEGG) gene sets were used to analyze related enriched biological functions and pathways [44].

### 2.7. Weighted Gene Coexpression Network Analysis 

The top 50% of the most variant genes were selected to construct a WGCNA by applying the “WGCNA” package of the R software [45]. After the optimal soft-threshold β was determined, the expression matrix was transformed into an adjacency matrix, and then into a topological overlap matrix. Based on the degree of dissimilarity between genes, the dynamic tree cut method was used to divide genes into different modules (the minimum number of genes in a module was 30). Modules with correlation greater than 0.8 were merged. Correlation coefficients and *p*-values between each module eigengenes and traits (high- and low-risk groups) were calculated. The module eigengene most related to the low-risk group was determined, and the genes within the module were considered to be low-risk group related genes. Finally, Metascape was performed to annotate and analyze low-risk group related genes [46].

### 2.8. Correlation between Risk Model and Immune Cell Infiltration Patterns 

CIBERSORT algorithm and single-sample GSEA (ssGSEA) were used to calculate the infiltration ratio and enrichment fraction of tumor-infiltrating immune cells in patients with HNSCC in comparison to the immune cell infiltration between high- and low-risk groups. The correlation between risk score and model gene and tumor-infiltrating immune cells was calculated, respectively [47,48]. The “estimate” package was used to calculate the scores of stromal cells and tumor-infiltrating immune cells in high- and low-risk groups and to evaluate the interstitial score and immune score of high- and low-risk groups. The ESTIMATE score is a combination of these two scores. According to the ESTIMATE score, tumor purity can be further calculated [49].

### 2.9. Immune-Related Gene Analysis of Risk Model and Prediction of Immunotherapy Efficacy

At the same time, we also compared the target genes under research or potential targeted therapy in the treatment of HNSCC with the marker genes of immunotherapy between high- and low-risk groups [50].TCIA database is based on the TCGA database to analyze the immune data of 20 cancer species, and a quantitative score of immunophenoscore (IPS) is developed, which is a better predictor of immunotherapy response to cytotoxic T-lymphocyte-associated antigen 4 (CTLA-4) and programmed cell death 1 (PD-1) antibodies [51,52]. The IPS score downloaded from the TCIA database was compared between different risk groups as a therapeutic effect verification.

### 2.10. Small Interfering RNA Transfection

After SCC25 and CAL27 cells were seeded into 6-well plates and incubated for 24 h, 30 nM negative controls (si-NC) or si-RNAs were added per well according to the manufacturer’s instruction of Pepmute Transfection Reagent (Signagen, Rockville, MD, USA). RNA was extracted or biological function experiments were performed after 48 h. SiRNA sequences were listed in Appendix A. 

### 2.11. Quantitative Real-Time PCR (qRT-PCR)

The total RNA was extracted from cells and tissues by RNAzol® RT (American Center for Molecular Research). Then, 1 µg of total RNA was reverse-transcribed to complementary DNA using the Hifair III 1st Strand cDNA Synthesis kit (Yeasen, Shanghai, China). qPCR reactions were performed by LightCycler 480 II (Roche, Basel, Switzerland) using SYBR Green Master Mix (Yeasen, China) as previously described [53]. Gene expression was calculated by 2^-ΔΔCt^ normalized to β-ACTIN. The primer sequences used were listed in Appendix A.

### 2.12. Wound-Healing, Migration, and Invasion Assays

For wound-healing assays, SCC25 and CAL27 cells were cultivated in 6-well plates to a confluency of more than 90%. Sterile pipette tips were used to scrape the cells, and then the cells were washed twice with phosphate-buffered saline (PBS) and treated with FBS-free media. At a 40× magnification, three randomly selected fields of view were captured under a microscope. The methods for migration and invasion assays were described in previous studies [54].

### 2.13. Statistical Analysis

All in vitro experimental data were expressed as mean ± standard deviation. R software or GraphPad Prism 9.0 software was used to conduct statistical analysis. Using the Shapiro–Wilk test to determine the normal distribution of the data. Two-tailed unpaired or paired Student’s *t*-test was used to statistically analyze the two groups of normally distributed data; otherwise, the Mann–Whitney U test was used. One-way ANOVA was performed to statistically analyze three groups. A *p*-value < 0.05 was considered statistically significant.

## 3. Results

### 3.1. The Landscape of m^6^A Regulators in TCGA-HNSCC Cohort

The data used in this study were from the TCGA-HNSCC cohort, including 502 HNSCC samples and 44 normal samples. According to published articles, we obtained 26 m^6^A regulators expression matrices from the TCGA-HNSCC dataset. As shown in Figure 1A, we visualized the expression of each m^6^A regulator in HNSCC samples and normal samples using thermal maps. Compared with normal tissues, IGF2BP2, HNRNPC, YTHDF1, HNRNPA2B1, KIAA1429, RBM15, ELAVL1, FMR1, METTL3, IGF2BP3, HNRNPG, WTAP, IGF2BP1, METTL16, YTHDF2, CBLL1, ALKBH5, YTHDF3, METTL14, FTO, YTHDC1 were highly expressed in HNSCC tissues. With the criteria of *p*-value < 0.001, only YTHDC was not differentially expressed in normal tissues and tumor tissues. Similarly, compared with the paired normal tissues, except for YTHDC2, the m6A regulator was significantly highly expressed in HNSCC (*p* < 0.05) (Figure 1B). In the expression correlation analysis between m^6^A regulators, we were surprised to find a strong positive correlation between m^6^A regulators, indicating that these genes have a coexpression regulatory network (Figure 1C, Appendix A). The forest map showed that IGF2BP2, IGF2BP1, and HNRNPC were associated with poor prognosis in HNSCC patients, while their m^6^A regulators were not significantly associated with poor prognosis in HNSCC patients (Figure 1D).

### 3.2. Construction and Validation of the Risk Model Based on Prognostic m6A-Related eRNAs 

m^6^A regulator plays a post-transcriptional regulatory role in the malignant progression of HNSCC, and its relationship with tissue-specific eRNA in HNSCC patients has not been clarified. eRNA can act as a cis-or trans-acting element, or change the chromatin environment or interact with transcriptional regulators to regulate gene expression [55].M^6^A is involved in RNA metabolism, such as RNA splicing and stability, which may lead to dysregulation of eRNA expression. Based on this, the purpose of this study is to better understand the role of m^6^A-related eRNAs in prognosis prediction in HNSCC.

Among the 26 expression matrices of m^6^A regulators and 1580 annotated eRNA, Pearson correlation analysis showed that 120 eRNA were significantly correlated with m^6^A regulators (|*r*| > 0.4, *p* ≤ 0.001) (Figure 2A) (Appendix A). Univariate Cox regression analysis indicated that 32 m^6^A-related eRNAs were associated with OS in patients with HNSCC (Figure 2B). The TCGA-HNSCC cohort was randomly divided into a training set and a testing set according to the proportion of 7:3. In the training set, 12 genes (Figure 2C,D) closely related to the overall survival of HNSCC patients were obtained by LASSO regression analysis of 32 m^6^A-related eRNAs. The 12 m^6^A-related eRNAs were further analyzed by multivariate Cox regression analysis to obtain a risk model consisting of 5 genes. The corresponding coefficients were shown in Table 1, and the risk score of each HNSCC patient was calculated. We used a Sankey diagram to show the relationship between m^6^A regulators and risk model eRNA (Figure 2E). In vitro experiments suggested that knockdown of YTHDF2, YTHDC2, FTO, and HNRNPC could up-regulate or decrease the expression of MIAT, MIR44352HG, AC005562.1, COPDA1, and LINC00271, indicating that the expression of these 5 m^6^A-related eRNAs was regulated by their corresponding m^6^A regulators (Figure 2F–I). Through the SRAMP website, all the 5 m^6^A-related eRNAs were predicted to have m^6^A sites, indicating that their expression can be regulated by the related m^6^A regulators in the form of m^6^A modification (Appendix A) [56]. We divided the training set into high- and low-risk groups using the median risk score as the cutoff value. K-M survival analysis and ROC analysis in the training set were performed to further evaluate the value of the risk model in predicting the prognosis of HNSCC patients. K-M survival analysis revealed that patients in the high-risk group had a significantly lower OS (Figure 2J), while the AUC values of the 1-, 3-, and 5-year OS of the ROC curves were 0.713, 0.641, and 0.600, respectively (Figure 2K). Then we further validated the risk model in the TCGA-HNSCC testing set, which revealed a noticeably worse OS in the high-risk group compared with the low-risk group. The risk model also showed a favorable predictive ability in the testing set, with AUC values of 0.697, 0.704, and 0.637 for 1-, 3-, and 5-year survival, respectively (Figure 2L,M).

### 3.3. Construction and Verification of a Nomogram

We conducted a multivariate Cox regression analysis to investigate the prognostic value of clinical indicators (age, sex, stage, grade, alcohol) and risk score in the TCGA-HNSCC cohort. As shown in Figure 3A, age (hazard ratio (HR) = 1.021, 95% confidence interval (CI) = 1.007–1.036, *p* = 0.003) and risk score (HR = 2.151, 95% CI = 1.701–2.718, *p* < 0.001) were high-risk factors for independent prognosis of HNSCC patients. There was no significant difference in sex (*p* = 0.081), stage (*p* = 0.063), grade (*p* = 0.234), and alcohol (*p* = 0.765). In order to predict the prognosis of HNSCC patients more intuitively, we combined the risk score with the clinical indicators of multivariate Cox regression analysis *p* < 0.10 to draw a nomogram in the training set to predict 1-, 3-, and 5-year OS. Each indicator had corresponding scores in the fractional axis according to its contribution to survival, and the total scores obtained by adding the scores corresponding to all indicators corresponded to the patient’s predicted 1-, 3-, and 5-year OS, respectively (Figure 3B). In order to evaluate the consistency and robustness of nomogram in predicting the prognosis of HNSCC patients, we drew the calibration curves of the training set and the testing set, showing that the predicted OS of the patients according to the predicted line chart was in suitable agreement with the actual results (Figure 3C,D). By plotting the AUC value of the ROC curve and calculating the C-index, it was confirmed that the line diagram had suitable prediction accuracy. The AUC values of 1-, 3-, and 5-year OS in the training set were 0.714, 0.661, and 0.606, respectively. The AUC values of 1-, 3-, and 5-year OS in the testing set were 0.762, 0.728, and 0.739, respectively. The C-index was 0.645 (95% CI = 0.597–0.694) in the training set, and 0.704 (95% CI = 0.640–0.768) in the testing set. Based on the above results, the chart had suitable consistency and robustness in predicting the OS of HNSCC patients.

### 3.4. Identification of the Risk Model-Associated Enriched Biological Functions and Pathways

In the TCGA-HNSCC cohort, we selected the hallmark gene set and KEGG gene set and utilized GSEA to understand risk model-associated enriched biological functions and pathways. Based on the adjusted *p*-value < 0.05 and FDR < 0.05, 29 characteristics of hallmark gene sets were enriched, mainly “allograft rejection”, “epithelial-mesenchymal transition (EMT)”, “glycolysis”, “G2M checkpoint”, and “hypoxia” (Figure 4A, Appendix A). There were 53 enrichment pathways on KEGG gene sets, such as “allograft rejection”, “epicardial thyroid disease”, “ECM receptor interaction”, “primary immunodeficiency”, and “focal adhesion” (Figure 4B, Appendix A). Based on the enrichment characteristics of the two gene sets, we found that compared with the low-risk group, the high-risk group was positively correlated with EMT, glycolysis, and hypoxia, which were the characteristics of malignant progression of tumors. Meanwhile, the high-risk group was negatively correlated with allograft rejection and immunodeficiency, indicating low immunity. Next, we selected a soft threshold of 12 to construct a weighted gene coexpression network in the TCGA-HNSCC data set to screen out the light green gene module most relevant to the risk model and used Metascape to annotate the genes in the module (Figure 4C,D, Appendix A). As shown in Figure 4E, for patients in the low-risk group, genes were mainly enriched in the biological functions of “B cell activation”, “adaptive immune response”, and “negative regulation of immune system process”, indicating active immunity. Collectively, we found that a risk model consisting of 5 m^6^A-related eRNAs may affect the prognosis of HNSCC patients by participating in tumor progression-related and immune-related biological functions and pathways.

### 3.5. Correlation between Risk Model and Immune Cell Infiltration Patterns in TCGA-HNSCC Cohort

The interaction of immune cells and tumor cells within the tumor micro-environment in HNSCC is a significant component influencing tumor progression and patient clinical prognosis [57]. In order to investigate the differences in immune cell infiltration between high-risk and low-risk groups, we further examined the association between the risk model and immune cell infiltration. The results of the CIBERSORT algorithm showed that the infiltration rate of naïve B cells, CD8^+^ T cells, activated CD4^+^ memory T cells, and resting mast cells in the high-risk group was lower (*p* < 0.001), while that of resting CD4^+^ memory T cells, M0 macrophages, activated dendritic cells, activated mast cells were higher (*p* < 0.001) (Figure 5A). As shown in Figure 5B, the expression of protective genes COPDA1 and MIAT was positively correlated with the invasion ratio of naive B cells, CD8^+^ T cells, and activated CD4^+^ memory T cells (*p* < 0.001), while the expression of risk gene MIR4435-2HG was on the contrary. The results of ssGSEA (Figure 5C) showed that compared with the low-risk group, in the high-risk group, cells related to immune function such as activated B cells activated CD4^+^ T cells, and activated CD8^+^ T cells (*p* < 0.001) were also decreased. Moreover, the risk score was significantly negatively associated with various immune cells, indicating that patients in the low-risk group benefited from more immune-infiltrating cells (Figure 5D). Similarly, using the “estimate” package to calculate the scores of stromal cells and tumor-infiltrating immune cells in each patient in the TCGA-HNSCC cohort, the scores of multiple immune cells decreased in high-risk group patients, and the immune score and ESTIMATE score were also significantly lower than low-risk group (Figure 5E,F). In addition, the tumor purity of HNSCC patients in the high-risk group was also significantly higher than that in the low-risk group (Figure 5G).

### 3.6. Immune-Related Gene Analysis of Risk Model and Prediction of Immunotherapy Efficacy in TCGA-HNSCC Cohort

Immunotherapy, including adoptive cell transfer and immune checkpoint inhibitors (ICIs), has been shown to have significant therapeutic effects on a variety of cancers. However, due to the difference in the immune micro-environment in patients, only some patients benefit clinically. Therefore, understanding the immune micro-environment and checkpoints of each patient could contribute to alleviating immune tolerance and improving the effect of treatment [58]. We further explored the correlation of risk models with immune checkpoints and target genes to predict the efficacy of risk model subgroups for immunotherapy and targeted therapy. According to the differential expression results between high- and low-risk groups, in the high-risk group, CD274, CD40LG, CTLA-4, PDCD1, PDCD1L2, TLR8, TNFRSF4, and TNFTSF9 were significantly down-expressed, while the expressions of CCND1, EGFR, EPHA2, HRAS, IGF1R, MET, MYC, NF1, PIK3CA, and PTEN were significantly up-regulated (Figure 6A,B). Currently, ICIs widely used in HNSCC patients mainly act on PD-L1 (CD274) and CTLA-4, while targeted drugs mainly act on EGFR. According to our findings, the high-risk group’s EGFR expression was higher than that of the low-risk group, while CTLA-4 expression was lower. However, there was no significant difference in CD274 expression statistically. The TCIA-HNSCC data were separated into high- and low-risk groups by the cutoff value of the risk score. In the low-risk group, significantly higher IPS scores were shown in the CTLA-4+_PD-1+ group and the CTLA-4+_PD-1 group, and the two groups had the same p-value (Figure 6C). The combination of the above results predicts that anti-CTLA-4 therapy alone or a combination of anti-CTLA-4 and anti-PD-1 therapy may have similar effects in the low-risk group.

### 3.7. MIR4435-2HG Was Highly Expressed in HNSCC Tissues and Promoted Cell Migration and Invasion

By analyzing the expression differences of 5 m^6^A-related eRNAs in 43 pairs of samples from the TCGA-HNSCC cohort and 20 pairs of HNSCC samples from our hospital, we found that compared with normal tissues, the protective gene LINC00271 was lowly expressed in HNSCC samples, the protective gene MIAT was highly expressed in HNSCC samples due to compensatory elevation, and MIR4435-2HG was highly expressed as a risk gene in HNSCC samples (Figure 7A,B). Next, the migration and invasion of MIR4435-2HG in HNSCC cells were further investigated after the expression of MIR4435-2HG was silenced in SCC25 and CAL27 cell lines (Figure 7C). Compared with the si-NC group, silencing MIR4435-2HG inhibited the wound-healing ability of HNSCC cells (Figure 7D). Moreover, the transwell assay also proved that the migration and invasion abilities of SCC25 and CAL27 cells after MIR4435-2HG knockdown were significantly reduced compared with the si-NC group (Figure 7E,F).

## 4. Discussion

Due to the different clinical symptoms of HNSCC patients with different primary sites, there are still many patients who are initially diagnosed with advanced disease, often with cervical lymph node metastasis or even distant metastasis, resulting in a poor prognosis. Finding more effective and highly specific prognostic factors is conducive to guiding the formulation of reasonable treatment plans based on the prognostic outcomes of patients [2,4]. m^6^A is the most abundant modification of RNA, and recent studies have demonstrated a large deposition of m^6^A in newborn eRNA. The m^6^A modification on eRNA is associated with the characteristics of active enhancers, suggesting that m^6^A plays a key role in driving eRNA [39]. The production of eRNA is tissue specific, which is related to the function of enhancers in the response of many cell types to stimuli. eRNA is involved in the dysregulation of oncogenes and tumor suppressor genes, as well as abnormal responses of cells to a variety of external stimuli such as inflammation, hypoxia, and hormones [36]. At present, the role of m^6^A-related eRNA in tumorigenesis and progression has not been reported. Therefore, we believe that the study of the role of m^6^A-related eRNAs in HNSCC is helpful in discovering new biomarkers.

In this study, 120 m^6^A-related eRNAs were screened by coexpression analysis based on the TCGA-HNSCC cohort. Then, we identified 32 prognostic m6A-related eRNAs through univariate Cox regression analysis. TCGA-HNSCC cohort was randomly split into a training set and a testing set, with a proportion of 7:3. Subsequently, the risk model based on 5 eRNAs was identified in the training set, using LASSO regression analysis and multivariate Cox regression analysis. The model genes included AC005562.1, COPDA1, LINC00271, MIAT, and MIR4435-2HG. K-M survival analysis and ROC curves demonstrated that the risk model was highly correlated with OS of HNSCC patients with a suitable predictive effect in the TCGA-HNSCC training set and testing set. Moreover, it was predicted that all the 5 m^6^A-related eRNAs had m^6^A sites through the SRAMP website. After knocking down the related m^6^A regulators, the expression levels of MIAT, MIR4435-2HG, AC005562.1, COPDA1, and LINC00271 were significantly changed, indicating that their expression may be regulated by m^6^A modification of the related m^6^A regulators, but the specific molecular mechanism needs to be further explored and verified by experiments in the future.

Among these m^6^A-related eRNAs in the risk model, MIAT has been reported as a carcinogenic agent in a variety of tumors (such as gastric cancer, hepatocellular carcinoma, colorectal cancer, ovarian cancer, breast cancer, etc.). The high expression of MIAT is related to the clinicopathological characteristics of tumor patients. MIAT can act as ceRNA to regulate a variety of signal pathways in the cytoplasm and participate in the regulation of cancer cell proliferation, cycle, migration, invasion, and drug resistance [59]. However, the role of MIAT in head and neck squamous cell carcinoma has not been reported. In this study, we used TCGA-HNSCC datasets and clinical samples to verify the high expression of MIAT in HNSCC tissues, but the high expression of MIAT was beneficial to the prognosis of patients, which may be a compensatory increase in tumor to increase the infiltration of tumor immune cells and improve the anti-tumor immunity of HNSCC patients. MIR4435-2HG has also been shown to be up-regulated in gastric cancer, hepatocellular cancer, colorectal cancer, ovarian carcinoma, and renal cell carcinoma, which can promote the expression of downstream target genes through sponging miRNAs, inhibit tumor cell apoptosis, and enhance cell proliferation, invasion, migration, and EMT [60]. Wang et al. found that mir4435-2Hg was highly expressed in HNSCC tissues, and in vitro and in vivo experiments verified that MIR4435-2HG promoted HNSCC cell proliferation, migration, invasion, and tumor growth by regulating the miR-383-5P/RBM3 axis [61]. In this study, we also proved that MIR4435-2HG was highly expressed in HNSCC tissues, and in vitro experiments confirmed that silencing MIR4435-2HG can reduce the migration and invasion ability of HNSCC cells. AC005562.1, COPDA1, and LINC00271 have not been studied in tumors, and their role in tumors, especially HNSCC, needs to be further understood.

We performed GSEA using the TCGA-HNSCC dataset to identify biological functions and pathways associated with the risk model. The results of GSEA showed that the high-risk group was mainly related to genes sets such as allograft rejection, primary immunodeficiency, EMT, glycolysis, and hypoxia, which were associated with poor immune monitoring function, tumor cell metastasis, and metabolic reprogramming, resulting in poor prognosis of HNSCC patients. eRNAs may participate in a variety of cancer signal pathways and affect drug response by regulating target genes or immune checkpoints. For example, overexpression of eRNA NET1e in breast cancer promotes tumor growth and enhances the resistance of breast cancer cells to anticancer compounds [62]. By annotating the genes in the module most related to the risk model screened by WGCNA, we found that the suitable prognosis of the low-risk group was related to the pathways that activate immunity, such as B cell activation, adaptive immune response, and negative regulation of the immune system process. Recent studies have shown that the cell type specificity of eRNA (such as CD8^+^ T cells) contributes to the understanding of heterogeneity within tumors the heterogeneity within the tumor, which better explains the cancer phenotype than mRNA, and eRNA may regulate the immune checkpoint of melanoma patients treated with anti-PD-1 [63]. Additionally, various types of tumor-infiltrating immune cells may express eRNA differently. The relationship between eRNA and tumor-infiltrating immune cells has significant implications for the identification of eRNA-based immunotherapy biomarkers. Patients in the high-risk group in this study had considerably lower immune scores and ESTIMATE scores than those in the low-risk group. This difference can be ascribed to a decline in the number of immune cells, primarily activated B cells, active CD4^+^ T cells, and activated CD8^+^ T cells. The primary function of infiltrating B cells in anti-tumor immunity is their capacity to deliver antigens to CD4^+^ and CD8^+^ T cells, thus forming an antigen-specific immune response in the tumor micro-environment. Furthermore, B lymphocytes play a role in the development of tertiary lymphoid structures (TLSs). TLSs aid in the further maturation and subtype transformation of tumor-specific B cells, as well as the establishment of tumor-specific T cell responses [64]. CD4^+^ T cells can bind to the non-polypeptide domain of MHC class II molecules, take part in T cell antigen receptor (TCR) signal transduction, and then initiate the differentiation of CD8^+^ T cells into cytotoxic T lymphocytes (CTL), maintaining and enhancing CTL’s anti-tumor response. On the other hand, CD4^+^ T cells have the ability to directly kill tumor cells by IFN- even in the absence of CD8^+^ T cells, indicating that CD4^+^ T cells are essential for tumor immunity [65]. CD8^+^ T cells play a vital role in tumor immunity by amplifying and differentiating into CTLS that specifically kill tumor cells [62]. In addition, activated CD8^+^ T cells can generate a variety of cytokines (including IFN-γ, TNF-α, and lymphotoxin -α) that indirectly induce tumor cell death [66]. Tumor purity, referring to the proportion of tumor cells in the tumor tissue, can be evaluated by the ESTIMATE algorithm, and the ESTIMATE score is negatively correlated with tumor purity. Tumor tissues are frequently mixed with non-tumor cells, such as stromal and immune cells, which play vital roles in tumor growth, progression, and drug resistance [57]. As shown in Figure 5E–G, patients in the high-risk group had higher tumor purity and lower immune score when compared with those in the low-risk group while showing no statistically significant difference in the stromal score. This further indicated that HNSCC patients with low tumor purity had various immune cell infiltration, which improved patients’ prognosis. The above results showed that patients in the high-risk group may have a worse prognosis due to fewer tumor-infiltrating immune cells, anti-tumor immune suppression, and local adaptive immune dysfunction.

Tumor immunotherapy and targeted therapy are two promising new anti-tumor therapy methods. Immunotherapy can improve the clinical prognosis of patients by activating anti-tumor immune responses and killing cancer cells [67]. Targeted therapy is targeted at specific targets on cancer cells and is an accurate treatment for tumors by interfering with the growth, division, and spread of cancer cells [68]. We further analyzed the correlation between high- and low-risk groups and immune genes or target genes that can be applied to HNSCC treatment and found that the expression of multiple immune genes was decreased in patients in the high-risk group, while the expression of multiple target genes was significantly up-regulated in the low-risk group. At present, ICIs, which are widely used in HNSCC patients, mainly act on PD-1 (PDCD1)/PD-L1 (CD274) and CTLA-4, while targeted therapeutic drugs mainly act on EGFR [69,70]. Our results showed that compared with the low-risk group, the expression of CTLA-4 and PDCD1 decreased, and the expression of EGFR increased in high-risk group patients, but there was no statistical difference in CD274. It was shown that targeted therapy was more effective in the high-risk group, while immunotherapy may be more beneficial in the low-risk group. The TCIA-HNSCC data further predicted that anti-CTLA-4 alone or in combination with anti-CTLA-4 and anti-PD-1 therapy may be more effective in low-risk group patients.

However, our research still has some limitations. First of all, the lack of independent validation of external datasets requires further validation of the risk model and nomogram by multi-center prospective studies. Second, the clinical data of the TCGA-HNSCC cohort is incomplete, and some potential risk factors, such as HPV infection and adverse pathological features, are not included in the nomogram. Third, more studies are needed to verify the specific mechanism of m^6^A and eRNAs in the occurrence and development of HNSCC. 

## 5. Conclusions

In summary, our study screened and identified a risk model containing 5 m^6^A-related eRNAs in the m^6^A regulators and eRNAs expression profile of the TCGA-HNSCC cohort, which can predict the survival and immunotherapy outcome of patients. We also combined the risk score and multiple clinical features to construct a nomogram for predicting the OS of HNSCC patients, which provides an effective quantitative analysis tool for patients’ personalized and precise treatment.

## Figures and Tables

**Figure 1 cancers-14-04657-f001:**
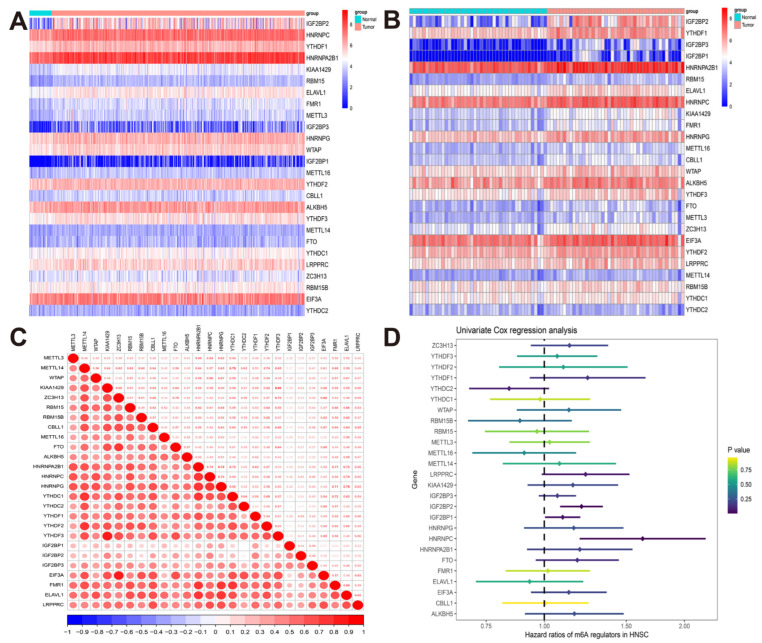
The landscape of m^6^A regulators in TCGA-HNSCC cohort. (**A**) Heatmap showing the expressions of m^6^A regulators between 44 normal and 502 tumor tissues; (**B**) heatmap showing the expressions of m6A regulators between 43 tumors and paired normal tissues; (**C**) the correlation between the m^6^A regulators in the TCGA-HNSCC cohort; (**D**) forest plot of univariate Cox regression analysis of m^6^A regulators associated with OS.

**Figure 2 cancers-14-04657-f002:**
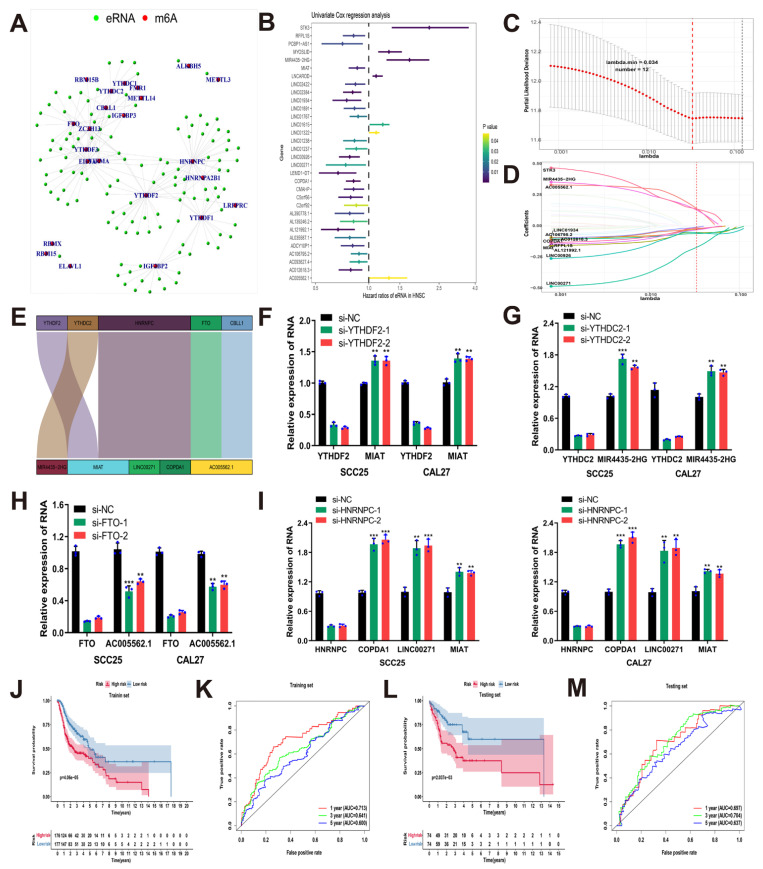
Construction and validation of the risk model based on prognostic m^6^A-related eRNAs. (**A**) Network of m^6^A regulators as red nodes and 120 eRNAs as green nodes; (**B**) forest plot of univariate Cox regression analysis of 32 m^6^A-related eRNAs associated with OS; (**C**) selection of the optimal lambda in the LASSO model; (**D**) LASSO coefficient of the 12 m^6^A-related eRNAs; (**E**) Sankey diagram showing the relationship between m^6^A regulators and m^6^A-related eRNAs; (**F**) RT-qPCR were used to detect MIAT expression after transfection of siYTHDF2 in SCC25 and CAL27 cells; (**G**) RT-qPCR were used to detect MIR4435-2HG expression after transfection of si-YTHDC2 in SCC25 and CAL27 cells; (**H**) RT-qPCR were used to detect AC005562.1 expression after transfection of si-FTO in SCC25 and CAL27 cells; (**I**) RT-qPCR were used to detect the expression of COPDA1, LINC00271, MIAT after transfection of si-HNRNPC in SCC25 and CAL27 cells; (**J**) Kaplan–Meier plots showing the difference between OS in low and high-risk groups in training set; (**K**) the ROC curves and AUC values of the risk model for 1-, 3-, and 5-year survival in training set; (**L**) Kaplan–Meier plots showing the difference between OS in low and high-risk groups in testing set; (**M**) the ROC curves and AUC values of the risk model for 1-, 3-, and 5-year OS in testing set. (** *p* < 0.01, *** *p* < 0.001).

**Figure 3 cancers-14-04657-f003:**
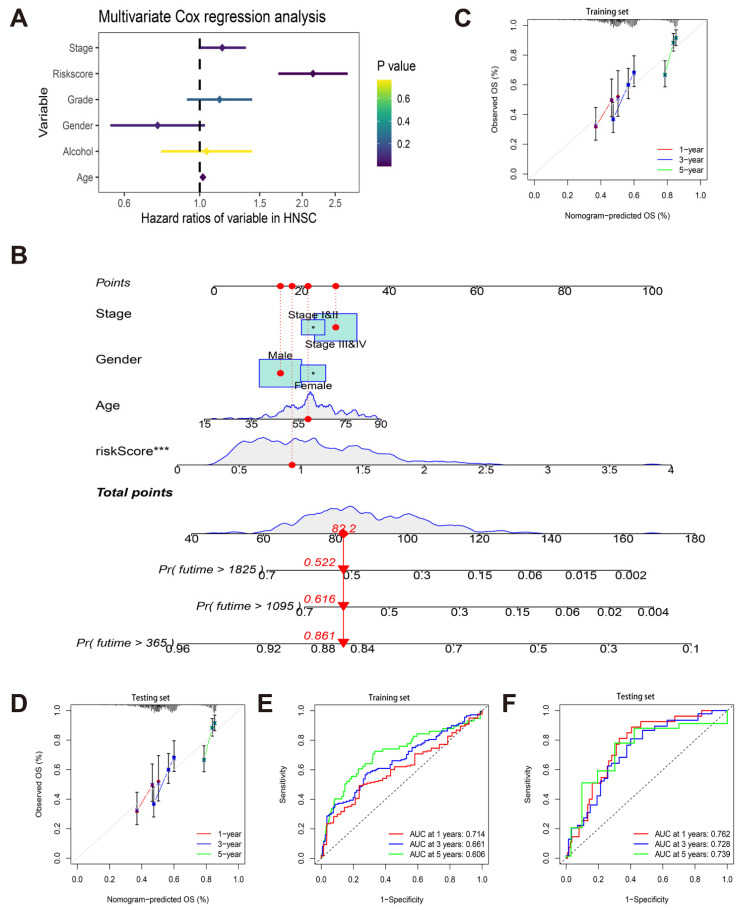
Construction and verification of a nomogram. (**A**) Forest plots of multivariate Cox regression analysis involving the risk score and clinical parameters; (**B**) prognostic nomogram for predicting OS of HNSCC patients based on TCGA training set; (**C**,**D**) the nomogram calibration curves for predicting OS at the time point of 1, 3, and 5 years in the training set (**C**) and testing set (**D**); (**E**,**F**) the ROC curves and AUC values of the nomogram for 1-, 3-, and 5-year OS in the training set (**E**) and testing set (**F**). (*** *p* < 0.001, The *p* value of multivariate Cox regression analysis of all indicators included in the nomogram).

**Figure 4 cancers-14-04657-f004:**
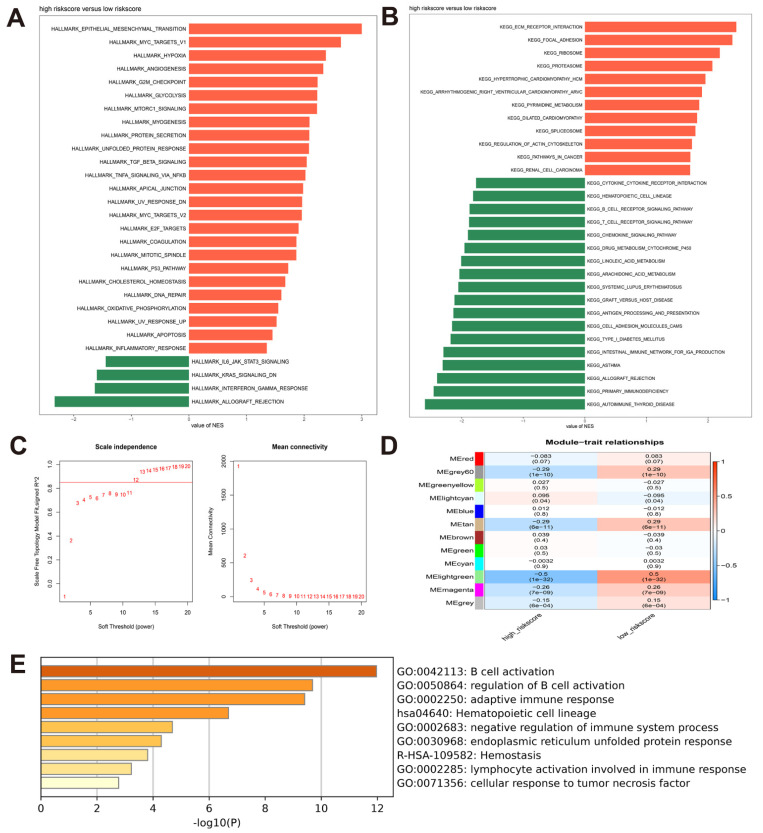
Identification of the risk model-associated enriched biological functions and pathways. (**A**) GSEA analysis of the enriched hallmark gene sets in the high-risk group; (**B**) GSEA analysis of the enriched KEGG gene sets in the high-risk group; (**C**) analysis of scale-free gene network topology and mean connectivity of various soft-threshold powers in TCGA-HNSCC cohort; (**D**) the correlation coefficients and p-values of module-risk relationships for TCGA-HNSCC cohort; (**E**) functional annotation of genes within the light green module.

**Figure 5 cancers-14-04657-f005:**
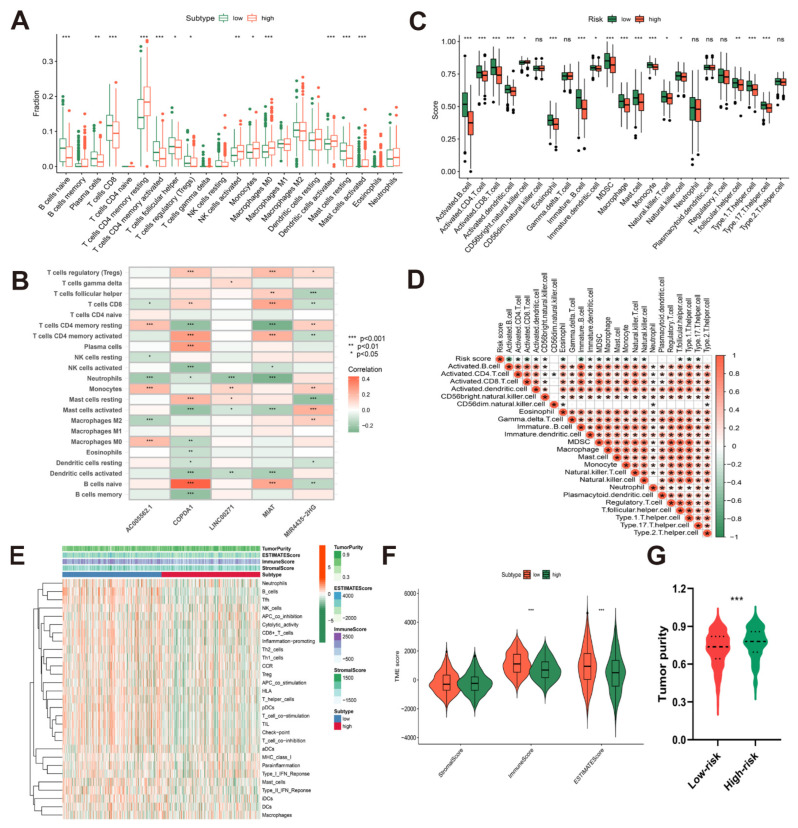
Correlation between risk model and immune cell infiltration patterns in TCGA-HNSCC cohort. (**A**) Infiltration fraction of high- and low-risk groups was assessed using CIBERSORT algorism; (**B**) the correlation analyses between the 5 m^6^A-related eRNAs of the risk model and immune cells; (**C**) infiltration score of high- and low-risk groups was evaluated using ssGSEA algorism; (**D**) the correlation analysis between risk score and immune cells; (**E**) heatmap showing the tumor purity, ESTIMATE score, immune score, stromal score, and immune cells infiltration score of high- and low-risk groups; (**F**) violin plot revealing the differences in ESTIMATE score, immune score, stromal score between high- and low-risk groups; (**G**) violin plot revealing the differences in tumor purity between high- and low-risk groups. (* *p* < 0.05, ** *p* < 0.01, *** *p* < 0.001).

**Figure 6 cancers-14-04657-f006:**
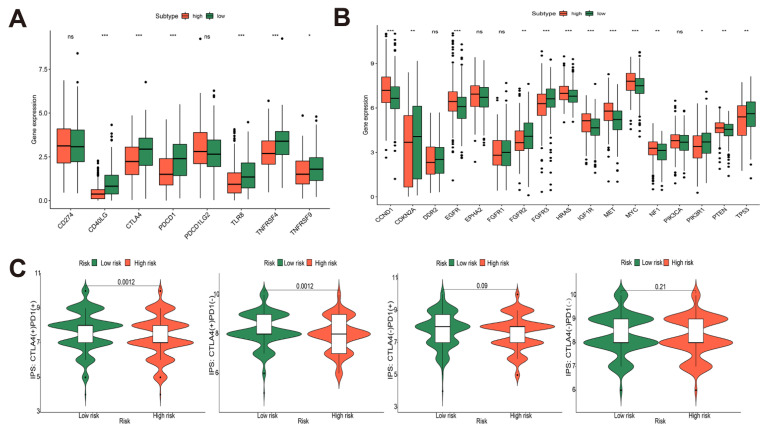
Immune-related gene analysis of risk model and prediction of immunotherapy efficacy in TCGA-HNSCC cohort. (**A**,**B**) The expression of immune checkpoints (**A**) and targeted genes (**B**) in high- and low-risk groups; (**C**) the IPS was compared in high- and low-risk groups. (* *p* < 0.05, ** *p* < 0.01, *** *p* < 0.001).

**Figure 7 cancers-14-04657-f007:**
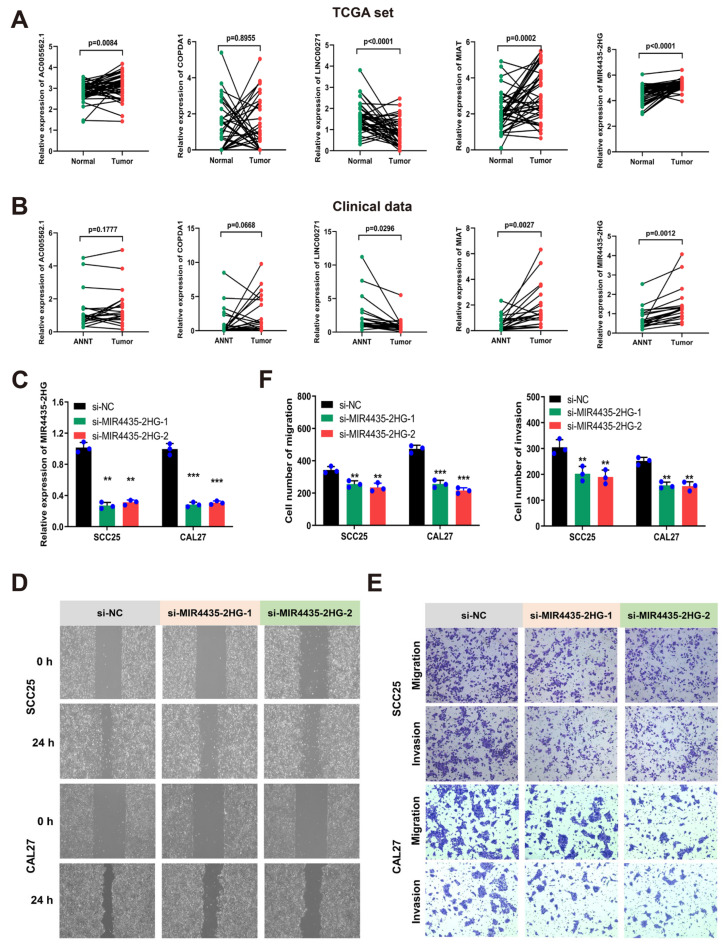
MIR4435-2HG was highly expressed in HNSCC tissues and promoted cell migration and invasion. (**A**) The expression of 5 m^6^A-related eRNAs between 43 tumor tissues and paired normal tissues in the TCGA-HNSCC cohort; (**B**) the expression of 5 m^6^A-related eRNAs between 20 HNSCC tissues and paired ANNT; (**C**) RT-qPCR analysis of SCC25 and CAL27 cells after transfection with si-MIR4435-2HG; (**D**) wound-healing assay images of SCC25 and CAL27 cells after transfection with si-MIR4435-2HG; Magnification at 40×; (**E**,**F**) representative photographs (**E**) and plots of statistical analysis (**F**) of migration and invasion assays of SCC25 and CAL27 cells after transfection with si-MIR4435-2HG; Magnification at 100×. The experiment was repeated three times; error bars indicate SD. (** *p* < 0.01, *** *p* < 0.001).

**Table 1 cancers-14-04657-t001:** Results of multivariate cox regression analysis of 5 m6A-related eRNAs and their coefficients.

Id	Coefficient	HR	HR.95 Low	HR.95 High	*p*-Value
MIR4435-2HG	0.354	1.425	1.015	2.002	0.041
LINC00271	−0.395	0.674	0.491	0.924	0.014
COPDA1	−0.113	0.893	0.792	1.006	0.063
AC005562.1	0.391	1.478	1.118	1.955	0.006
MIAT	−0.131	0.878	0.765	1.007	0.062

HR: hazard ratio.

## Data Availability

All the data are available from the TCGA database.

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
