# Peer review of "Integrative Analysis of N6-Methyladenosine-Related Enhancer RNAs Identifies Distinct Prognosis and Tumor Immune Micro-Environment Patterns in Head and Neck Squamous Cell Carcinoma"

_cancers, 2022, doi:10.3390/cancers14194657_

Round 1

Reviewer 1 Report

This article investigated TCGA-HNSCC data for the role of enhancer RNA and validated with their tissue

Major point

1. Estimate score is positively correlated with tumor purity. The author showed that Estimate score was lower in the high risk group. Please add the biologic meaning for this in the discussion.

2. The authors argued that they can predict the immunotherapy efficacy in HNSCC with their model. However there have been reported several makers for immunotherapy including TMB, MSI, dMMR and blood neutrophil to lymphocyte and so on. In addition TCGA data did not have the data for response of immunotherapy. TCIA provides immunophenoscore, which was not validated with external data. Based on these findings, the results and discussion for immunotherapy efficacy in the articel should be revised, or further clinical data for response of immunotherapy in HNSCC and risk score should be added.

Minor points

1. line 393

MIR4435-2Hg  

Author Response

Dera reviewer

We thank you very much for your time and effort in reviewing our paper (cancers-1894350). My co-authors and I appreciated the insightful comments and constructive critiques from you. 

Reviewer 2 Report

The manuscript titled: “Integrative analysis of N6-methyladenosine-related enhancer RNAs identifies distinct prognosis and tumor immune microenvironment patterns in head and neck squamous cell carcinoma” is well written. The manuscript is based on a well-constructed scientific concept and carried out the studies very well. I have no comments on the current manuscript. I would suggest accepting the manuscript in its present form.

Author Response

Dear reviewer

We thank you very much for your time and effort in reviewing our paper (cancers-1894350). My co-authors and I appreciated your acknowledgment and affirmation of our paper.